# Characterization of the Duodenal Mucosal Microbiome in Obese Adult Subjects by 16S rRNA Sequencing

**DOI:** 10.3390/microorganisms8040485

**Published:** 2020-03-29

**Authors:** Carmela Nardelli, Ilaria Granata, Valeria D'Argenio, Salvatore Tramontano, Debora Compare, Mario Rosario Guarracino, Gerardo Nardone, Vincenzo Pilone, Lucia Sacchetti

**Affiliations:** 1Department of Molecular Medicine and Medical Biotechnologies, University of Naples Federico II, 80131 Naples, Italy; carmela.nardelli@unina.it; 2CEINGE Biotecnologie Avanzate S. C. a R. L., 80131 Naples, Italy; dargenio@ceinge.unina.it; 3Task Force on Microbiome Studies, University of Naples Federico II, 80100 Naples, Italy; 4Institute for High Performance Computing and Networking (ICAR), National Research Council (CNR), 80131 Naples, Italy; ilaria.granata@icar.cnr.it (I.G.); mario.guarracino@unicas.it (M.R.G.); 5Department of Human Sciences and Promotion of the Quality of Life, San Raffaele Open University, 00166 Rome, Italy; 6Department of Medicine and Surgery, University of Salerno, 84084 Salerno, Italy; salvytra@libero.it (S.T.); vpilone@unisa.it (V.P.); 7Department of Clinical Medicine and Surgery, University of Naples Federico II, 80131 Naples, Italy; debora.compare@unina.it (D.C.); gerardoantoniopio.nardone@unina.it (G.N.); 8Department of Economics and Law, University of Cassino and Southern Lazio, 03043 Cassino, Italy

**Keywords:** obesity, duodenum, microbiome

## Abstract

The gut microbiota may have an impact on obesity. To date, the majority of studies in obese patients reported microbiota composition in stool samples. The aim of this study was to investigate the duodenal mucosa dysbiosis in adult obese individuals from Campania, a region in Italy with a very high percentage of obese people, to highlight microbial taxa likely associated with obesity. Duodenum biopsies were taken during upper gastrointestinal endoscopy in 19 obese (OB) and 16 lean control subjects (CO) and microbiome studied by 16S rRNA gene sequencing. Duodenal microbiome in our groups consisted of six phyla: Proteobacteria, Firmicutes, Actinobacteria, Fusobacteria, Bacteroidetes and Acidobacteria. Proteobacteria (51.1% vs. 40.1%) and Firmicutes (33.6% vs. 44.9%) were significantly (*p* < 0.05) more and less abundant in OB compared with CO, respectively. *Oribacterium asaccharolyticum*, *Atopobium parvulum* and *Fusobacterium nucleatum* were reduced (*p* < 0.01) and Pseudomonadales were increased (*p* < 0.05) in OB compared with CO. Receiver operating characteristic curve analysis showed Atopobium and Oribacterium genera able to discriminate with accuracy (power = 75% and 78%, respectively) OB from CO. In conclusion, increased Proteobacteria and decreased Firmicutes (Lachnospiraceae) characterized the duodenal microbiome of obese subjects. These data direct to further studies to evaluate the functional role of the dysbiotic-obese-associated signature.

## 1. Introduction

Obesity is an increasing worldwide health problem that is associated with several metabolic diseases [1]. In particular, among the Italian regions, Campania is one of those with highest presence of obese individuals (~14%), as measured by body mass index (BMI) > 30 kg/m^2^ (Data from https://www.epicentro.iss.it/passi/dati/sovrappeso). Except for rare forms of monogenic obesity, common obesity is a multifactorial disorder whose onset genetics, epigenetics, behavioral and environmental aspects contribute, from prenatal to adulthood [2,3,4,5,6,7]. A further factor likely to be involved in the insurgence of obesity is represented by the gut microbiota [8,9]. In humans, the bowel contains trillions of bacteria belonging to more than 1000 different species that represent a potential genetic space of more than three million bacterial genes [10]. The microbiome (the totality of all the genomic elements of a specific microbiota) retains a high degree of plasticity and its composition changes adaptively with age, diet and the use of medications as well as in relation to the healthy status of the host in a bi-directional way [11,12], even if the mechanisms underlying are still poorly understood. Moreover, it is very difficult to characterize the composition of the human gut microbiota due to large variations among individuals. Accumulating evidence suggest that composition and functions of gut microbiota differ between healthy lean subjects and obese patients and microbiota likely may have an impact on diseases associated with obesity, such as insulin resistance, low-grade inflammation, diabetes, and fatty liver disease [13,14,15]. The mechanisms, so far known, through which the gut microbiota is linked to obesity include energy extraction capacity from food, influence on the integrity of the gut barrier, modulation of the immune system and production of specific metabolites such as short-chain fatty acids, secondary bile acids, trimethyl-amine-N-oxide, branched-chain amino acids [14,16]. To date, the vast majority of the studies conducted in obese patients report microbiota composition of fecal samples [17] and few data are available regarding the characterization and the functional activity of microbiota especially at level of small intestine, in particular in duodenum [15,18,19].

It is to underline that bacterial composition differs in the various parts of the human gastrointestinal tract, depending on pH, oxygen content, host secretion, substrate availability and transit times of the content, resulting in a progressive decrease of aerobes and increase of obligate anaerobes from the proximal to the distal end of the gastrointestinal tract [20]. In particular, as previously demonstrated by 16S rRNA gene sequencing analysis, the bacterial community in small intestine is remarkably different from that in colon [21]. Further, the human small intestine is responsible for a large part of nutrient (carbohydrates, proteins and fats) digestion and absorption, amounting to 10%-30% of the total diet energy that the colon receives (indigestive carbohydrates and proteins) [22].

The aim of this study was to investigate the obesity-associated duodenal microbiome by 16S ribosomal RNA (rRNA) sequencing in adult obese and normal weight control individuals from Campania (Italy), to highlight microbial taxa associated to obesity and/or to obese-associated metabolic imbalance.

## 2. Materials and Methods

### 2.1. Patients and Controls

We studied two groups of subjects living in the Campania region for at least two generations: A) nineteen obese (OB) patients, enrolled consecutively among those eligible for bariatric surgery, divided in moderately obese (OB-1), with BMI = 30–40 kg/m^2^ (*n* = 13, 54% females), aged 20–56 years and severely obese patients (OB-2), with BMI > 40 kg/m^2^ (*n* = 6, 66% females), aged 24–60 years; and B) sixteen normal weight controls (CO) suffering from typical gastroesophageal reflux disease symptoms, with BMI 20.0–24.9 kg/m^2^, (38% females), aged 35–80 years. All study groups underwent gastrointestinal endoscopy as part of their diagnostic path and were enrolled after the histological examination revealed a normal mucosa. Patients and controls were enrolled at the Surgery and Medicine Department of the “Università degli Studi di Salerno” and “Università degli Studi di Napoli Federico II”. The study was approved by the Ethics Committee of the Universities (“Università degli Studi di Salerno”: authorization n.50, 15/07/2015; amendment n. 141070, 26/11/2019 and “Università degli Studi di Napoli Federico II”: no. 193/06, October 25, 2006; amendment no. 193/06/ESES1, October 1, 2014). All the enrolled subjects gave their informed consent to participate in the study, carried out according to the Helsinki Declaration. Exclusion criteria were: diabetes, tumours, inflammatory bowel diseases, Crohn disease, viral hepatitis, any pharmacological treatment (i.e. antibiotics, pro- and pre-biotics, antiviral or corticosteroid medications for at least 2 months before the collection of samples). The clinical, anamnestic and dietary habit data of each subject were collected by an expert clinician and nutritionist, respectively. Metabolic syndrome (MS), namely the presence of abdominal obesity, dyslipidemia (hypertriglyceridemia and low HDL cholesterol level), elevated blood pressure and hyperglycemia, was evaluated in all enrolled subjects.

### 2.2. Sample Collection and Biochemical Investigations

In the present study we collected the following biological samples from all enrolled individuals: blood samples for biochemical investigations and one duodenal biopsy specimen. The biopsy sample was taken during upper gastrointestinal endoscopy performed within the diagnostic path, under sterile conditions to avoid contamination as detailed in Appendix A. Biopsies were immediately cooled in dry ice and stored at −80 °C until DNA isolation for microbiome analysis. Lipid and other main haematological parameters (Table 1) were evaluated by routine assays on ACHITECT i2000R System (Abbott Laboratories, Wiesbaden, Germany).

### 2.3. Microbial Sequencing-16S rRNA

DNA was extracted from duodenal samples using QIAamp DNA mini kit (Qiagen, Venlo, The Netherlands) from 16 controls, 13 OB-1 and 6 OB-2. All extractions were performed in a pre-PCR designated room. To deeply investigate the microbiome composition, we used a multiplexed 16S rDNA amplicon-based approach coupled with the NGS system MiSeq (Illumina, San Diego, CA, USA). In particular, 500 bp amplicons, spanning the V4-V6 hyper-variable regions of the 16S rRNA gene, were obtained. Each sample was individually amplified and purified (Agencourt AMPure XT beads, Beckman Coulter, Brea, CA, USA). Primers used in the first round of PCR contained the overhang sequences with Illumina adapters; forward primer, 5’- TCGTCGGCAGCGTCAGATGTG TATAAGAGACAGCCAGCAGCCGCGGTAAT-3’; reverse primer, 5’-GTCTCGTGGGCTC GG AG ATGTGTATAAGAGACAGGGGTTGCGCTCGTTGC- 3’. PCR conditions were 95 °C for 10 min; 45 cycles of 95 °C for 30 s, 59 °C for 30 s, and 72 for 1 min. A second round of PCR was used to add the Illumina index to the amplicons for the library preparation according to the Nextera XT protocol (Illumina). PCR conditions were 95 °C for 3 min; 8 cycles of 95 °C for 30 s, 55 °C for 30 s, and 72 for 30 s. After appropriate quality assessment (TapeStation, Agilent Technologies, Santa Clara, CA, USA), the amplification products from different DNA samples were pooled in equimolar ratios. The obtained multiple amplicon libraries were quality assessed (TapeStation, Agilent Technologies) and quantified (Qubit dsDNA BR assay, Thermo Fisher, Waltham, MA, USA), according to the manufacturer’s instructions, in order to obtain a pool of equimolar libraries, so ensuring a normalization across the different samples sequenced in the same run. All libraries were sequenced with the Illumina PE 300 MiSeq System protocol, according to the specifications of the manufacturer (see Appendix A and Methods—Sample Processing). 16S sequencing raw data have been submitted to the NCBI SRA repository under the accession number PRJNA611696.

### 2.4. Microbiome Data Processing

To analyze the taxonomic composition of samples, DADA2 v. 1.15.0 [23] and Phyloseq 1.28.0 [24] R packages were used. A scarce overlapping between paired-end reads was observed. In this case, two analysis strategies are commonly suggested: usage of only forward reads as single ends or concatenation of forward and reverse reads. In the latter method, DADA2 concatenates reads by inserting Ns between them. We chose to apply the first strategy to have a more reliable alignment and since the too many Ns prevent the species annotation. We further compared the results obtained by using forward reads with those obtained by using the script join_paired_ends.py from Qiime software [25] and combining the joined and un-joined reads. In the end, we decided to rely on the forward reads strategy considering that forward and reverse reads match to two different hypervariable regions characterized by a different specificity and that the significant different genera in groups comparison analysis were almost the same, except one (Megasphaera, *p*: 0.046) (See also Appendix A,— Data Processing).

Before aligning reads, the forward primer was trimmed out from reads and these were filtered according to the following parameters: maxEE = 2; minLen = 50; maxN = 0; truncQ = 2. In details, as described in the software manual, maxEE parameter determines that the reads with "expected errors" higher than maxEE will be discarded, where expected errors are calculated from the nominal definition of the quality score: EE = sum(10^(-Q/10)); minLen indicates the minimum length to keep the reads; maxN indicates the maximum number of Ns allowed; truncQ allows to truncate reads at the first instance of a quality score less than or equal to truncQ. The reads were then denoised through the core sample inference algorithm. The DADA2 algorithm inferred, on average, 199.9 true sequence variants from the unique sequences in the CO group, 180 in the OB-1 group and 181.5 in the OB-2 group. After chimeric sequences removal, taxonomy was assigned to Amplicon Sequence Variants (ASVs) by using the SILVA reference database v.128 formatted for DADA2 software and available at the link: https://zenodo.org/record/824551#.XmIcO5NKhuU [26].

The phylogenetic tree was constructed by performing a multiple alignment using the DECIPHER 2.12.0 R package [27]. The phangorn 2.5.5 R package [28] was then used to first construct a neighbour-joining tree, and then fit a Generalized time-reversible with Gamma rate variation (GTR+G+I) maximum likelihood tree using the neighbour-joining tree as a starting point.

Statistical analyses of the dataset were carried out through combining all the data (cleaned ASVs, taxa assignment, phylogenetic tree, and metadata) into a phyloseq object.

The richness has been estimated on originally-observed counts through three α-diversity measures (Chao-1, Shannon, Simpson). Wilcoxon rank-sum test (Mann-Whitney) was performed in R environment to test the significance of pairwise richness differences. The β-diversity has been evaluated through weighted and unweighted Unifrac metrics on variance stabilizing transformed data (DESeq2 1.24.0 R package), as previously suggested [29]. The ANOSIM test was performed by using the homonym function provided by the Vegan 2.5–6 R package. The significance of differential abundance between two (CO vs. OB) and three (CO vs. OB-1 vs. OB-2) groups at each taxonomic level was assessed by Kruskal-Wallis Rank Sum Test in R environment [30]. In the case of three groups, the pairwise comparison was then performed by Dunn’s test [31], through FSA 0.8.27 package, on the significant Kruskal-Wallis tests and the *p*-value corrected for multiple comparisons by the Benjamini-Hochberg adjustment method. The Area Under the ROC Curve (AUC) was calculated by the colAUC function of the caTools 1.18.0 R package [32] and values >0.70 were considered accurate in discriminating study groups.

### 2.5. Statistical Analysis

The parameters investigated were expressed as mean and standard error of the mean (SEM) (parametric distributions) or as median value and 25th and 75th percentiles (nonparametric distributions). The Student’s ‘t’ test and Mann–Whitney test were used to compare parametric and nonparametric data, respectively. *p* values < 0.05 were considered statistically significant. Correlation analysis was performed with the SPSS package for Windows (ver. 18; SPSS, Inc., Headquarters, Chicago, Il, USA).

## 3. Results

### 3.1. Hematological and Clinical Parameters of the Studied Groups

The clinical and biochemical characteristics of the enrolled obese individuals are reported in Table 1. Moderate and severe obese patients, showed a statistically significant difference in BMI and glucose level [mean (SEM), OB-1 = 36.0 (0.8) Kg/m^2^, OB-2 = 46.5 (2.0) Kg/m^2^, *p* < 0.001; OB-1 = 4.8 (0.1) mmol/L, OB-2 = 5.5 (0.2) mmol/L, *p* < 0.001]. We also observed in the OB-2 group a trend in increased systolic blood pressure, total cholesterol and triglycerides, respect to OB-1, even if at not significant level. The clinical and biochemical parameters of the normal weight controls were all within the reference intervals for healthy subjects (data not shown). Metabolic syndrome was present in 6/19 obese patients and absent in controls.

### 3.2. Duodenal 16S rRNA Analysis

Appendix A reports the total sequencing reads obtained from 35 duodenal mucosa samples, with the mean value of sequences respectively in CO, OB-1 and OB-2. To test the overall differences of microbial community structures in obese patients and controls, alpha diversity was measured by Chao1, Shannon and Simpson indices. All indexes suggested a trend of decreased richness in OB respect to CO, but no statistically significant differences were highlighted (Figure 1).

To assess the differences between microbial composition in obese patients and controls, beta diversity was evaluated by the unweighted and weighted Unifrac distances using PCoA ordination method (Figure 2A,B).

UniFrac is a β-diversity measure that uses phylogenetic information to compare environmental samples. The unweighted is a quality-based distance measure while the weighted is a quantitative based distance. By weighted UniFrac analysis, we highlighted significant difference among the three study groups (*p* = 0.039, R = 0.063), so confirming that the variation between groups is not due to the type of taxa present in the microbiome but to their abundances (Figure 2B).

Taxonomic assignment indicated that the duodenal microbiome in the obese and normal weight individuals consisted of 6 distinct phyla: Proteobacteria (CO = 40.1%, OB = 51.1%), Firmicutes (CO = 44.9%, OB = 33.6%), Actinobacteria (CO = 6.4%, OB = 5.7%), Fusobacteria (CO = 6.4%, OB = 5.5%), Bacteroidetes (CO = 0.8%, OB = 1.7%) and Acidobacteria (CO = 0.2%, OB = 1.2%) with a relative abundance >1% in at least 1/2 study groups (Figure 3A).

Proteobacteria and Firmicutes phyla showed significant (*p* < 0.05) higher and lower abundance in OB compared to CO, respectively (Figure 3A). Kruskal-Wallis differential analysis between the two groups showed a significant reduction in both Firmicutes and Actinobacteria bacteria from class up to genus level (*p* < 0.01) and a significant increase in Pseudomonadales (Proteobacteria) order (*p* < 0.05) in OB respect to CO (Figure 3B–E).

In particular, bacterial species significantly (*p* < 0.01) reduced in OB respect to CO were *Oribacterium asaccharolyticum* (Firmicutes), *Atopobium parvulum* (Actinobacteria) and *Fusobacterium nucleatum* (Fusobacteria) (Figure 3F).

No statistically significant difference in taxa abundance was observed in obese patients when they were divided according to obesity severity in OB-1 (moderately obese) and OB-2 (severely obese) (see upper right corner of Figure 3A–E).

To further assess the strength of the association between significant bacterial taxa and obesity we also calculated the AUROCs and the genera Atopobium and Oribacterium resulted able to discriminate with accuracy (power = 75% and 78%, respectively) the two groups of OB and CO (Figure 4).

## 4. Discussion

So far, obesity-associated gut dysbiosis has been prevalently investigated in stool samples, which are easy to obtain but whose microbiota is mostly representative of colon bacterial composition [14,15,16]. Unlike that, duodenum microbiome, also involved in nutrient digestion, has been little investigated in obesity, due to the difficulties in obtaining duodenal biopsy samples [18].

Here, we report the gut microbiome characterization by 16S rRNA gene sequencing of duodenal mucosa samples, obtained when endoscopy is part of the diagnostic path, in obese patients and normal weight controls, aimed to obtain information on any microbial alterations present in the small intestine in obesity.

Duodenal microbiome in our study groups consisted of six phyla among which Proteobacteria and Firmicutes were the most abundant with increased and decreased relative abundance, respectively, in obese patients compared to normal weight controls.

In agreement with our data, in a metanalysis of the obesity-associated gut microbiota alterations, the decrease in the absolute number of sequences of Firmicutes in obese subjects respect to lean controls was the only reproducible and significant alteration observed [33]. The Lachnospiraceae family significantly contributed to the decreased abundance of Firmicutes observed in our obese group. Lachnospiraceae have been described as short chain fatty acids (SCAFs) producers exerting a beneficial effect on the intestinal barrier [34]. Their abundance was also positively and negatively associated with dietary fat and carbohydrates, respectively [35].

In addition, in a recent report on the gut microbiota of elderly obese women living in Italy, in agreement with our data, a tendency to decreased biodiversity in obese compared with control fecal microbiotas was observed as well as a reduced proportion of a number of health-promoting SCAFs producers belonging to Lachnospiraceae [36]. Further, the same authors found a negative correlation between baseline abundance of Lachnospiraceae and BMI and waist circumference, but, after two weeks of hypocaloric Mediterranean diet the obesity-associated dysbiotic signatures were reversed [36]. Lachnospiraceae family includes potentially pathogenic bacteria found in stool microbiome in diabetes and obesity affected patients and significantly correlated to obesity parameters (waist circumference, BMI), systolic pressure and consumption of carbohydrates [37,38]. In line with the latter report, our obese group showed increased systolic blood pressure (>133 mmHg) and self-reported a dietary habit rich in carbohydrates.

The genera Stomatobaculum and Oribacterium, belonging to the Lachnospiraceae family, are both obligately anaerobic bacteria in the human oral cavity and were significantly reduced in our obese group compared to controls. This result is suggestive of a continuum of microbiome composition between mouth and duodenum, previously observed by our group in a different disease model [39]. In agreement, decreased Lachnospiraceae were previously described in oral microbiome from obese compared to non-obese type 2 diabetic individuals [40]. Characterization of *Stomatobaculum* and *Oribacterium asaccharolyticum*, highlighted that Stomatobaculum encoded in its genome the cysteine desulfurase gene [41], whereas the major metabolic end product of *Oribacterium asaccharolyticum* was acetate [42]. Interestingly, a recent study evaluated the associations of BMI with circulating microbiota biomarkers in African American men and found that propionic and butyric SCFAs, but not acetic acid, were significant positive predictors of BMI [43]. The significant obese-associated decreases in *Atopobium parvulum* (Actinobacteria) and *Fusobacterium nucleatum* (Fusobacteria), bacterial species previously associated to Crohn’s disease [44] and to tumorigenesis [45], respectively, remain unclear.

Finally, unnamed Pseudomonadales (Proteobacteria) were found significantly more present in our OB respect to CO. Members of this order include opportunistic gram-negative pathogens of clinical significance able in triggering innate immune response in the murine host [46], but the significance of Pseudomonadales presence in duodenum of obese patients deserves further investigation.

Angelakis et al. [18] previously reported microbiome composition in duodenal contents aspirated from 5 obese and 5 healthy subjects at 90’ after a solid-liquid meal, in the framework of a gastrointestinal lipolysis study. Overall, similar taxonomic profile was observed between obese and control subjects. Among the 11 phyla detected small but non significant differences were detected in Firmicutes (67% vs. 62%) and Proteobacteria (4% vs. 9.5%) in obese vs. control groups. Further, these authors reported an almost complete absence of Bacteroidetes (~0.2%). Different studied samples (duodenal content vs. mucosal content), sampling condition (after solid-liquid meal vs. fasting) and number of obese and control groups, make the comparison between Angelakis and our data unfeasible. Nevertheless, both studies highlight major differences in duodenal compared with distal gut microbiome, caused primarily by alterations in the abundances of the microbes present, such as the shift in the proportion of Firmicutes and Bacteroidetes phyla, rather than by the changes in their membership.

Further, the microbial Firmicutes/Bacteroidetes ratio is reported to be increased in obesity, but it is usually calculated in stool samples, whose microbiome is mostly representative of colon [47]. The latter microbiome is largely different from those present in other intestinal tracts such as duodenum or jejunum, which host lower than colon abundance of Bacteroidetes, probably related to a limited availability of mucin as carbon source for Bacteroidetes [18,21].

One limit of our study is the small number of the obese and normal weight subjects enrolled, mainly due to the difficulties in duodenum sampling. In particular, it’s difficult to collect human biopsies from “healthy” individuals. Considering this point, we considered that the microbiome of our “control” group including lean subjects with clinical symptoms of gastroesophageal reflux (GORD) could present any differences with that of healthy individuals. Despite this, biopsies from our controls were taken before diagnosis and prospective GORD therapy with proton pump inhibitors (PPI). Further, multiple oral bacteria that were reported in gut microbiome of PPI-users included increased genera Rothia, Enterococcus, Streptococcus, Sthaphylococcus and species Escherichia Coli [48], and levels of these bacteria did not differ among our obese and control groups.

Another limit of this study was the absence of negative and/or positive controls in the DNA extraction and processing. In general, for good scientific practice, controls should be included at all steps in microbiome studies [49]. In fact, as recently reported, DNA extraction kits and other laboratory reagents could be a source of contamination [50]. However, among contaminant taxa previously identified from multiple studies [50,51,52], Ralstonia and Rhizobiales were virtually absent, and Methylbacterium was present in low relative abundance in both our groups of Control and Obese subjects (<0.2 and <0.3, respectively). Concerning Pseudomonadales and Atopobium, other putative contaminant bacteria [50,51,52], these taxa were present in all subjects but with opposite mean relative abundances in our two groups, that are Pseudomonadales was increased in obese and decreased in control subjects (1.94 and 0.58, respectively), whereas Atopobium was decreased in obese and increased in control subjects (0.26 and 0.88, respectively). Overall, our results are consistent with an insignificant bacterial contamination during sample processing.

One strength of this study is that all the study groups belonged to a restricted geographical area, that is the Campania region, this feature reduced the inter-individual variability linked to different geographical areas and to different eating habits.

## 5. Conclusions

In conclusion, we first report the microbiome composition in duodenal mucosa of adult obese subjects. A significant increase in Proteobacteria and decrease in Lachnospiraceae (Firmicutes) characterized the microbiome of obese subjects. These data direct to further studies to evaluate the functional role of the dysbiotic obese-associated duodenal signature, also in relation to any modification in nutrient digestion and absorption likely concurring to obesity.

## Figures and Tables

**Figure 1 microorganisms-08-00485-f001:**
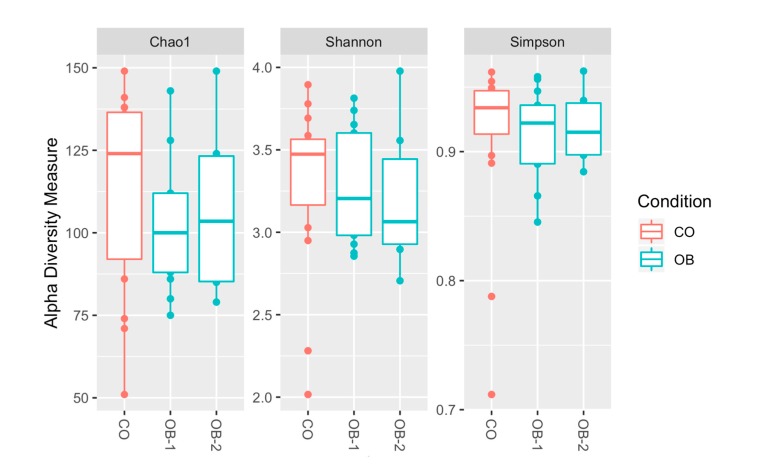
Alpha diversity of taxa identified in the Control (CO), Moderately Obese (OB-1) and Severely Obese (OB-2) groups. Alpha diversity analysis was performed through several metrics in order to assess the within-sample diversity and compare species richness between the different conditions under study. Chao1, Shannon entropy and Simpson diversity indices were calculated. Overall, the plots show a trend of decreased richness in OB-1 and OB-2 respect to CO, but no statistically significant differences were highlighted by performing the Wilcoxon rank-sum test (Mann-Whitney).

**Figure 2 microorganisms-08-00485-f002:**
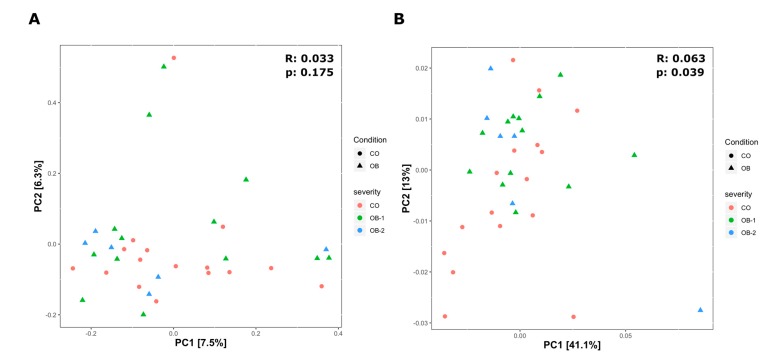
Beta diversity of bacteria identified in the Control (CO), Moderately Obese (OB-1) and Severely Obese (OB-2) groups. Principal coordinate analysis (PCoA) plots using the unweighted (**A**) and weighted (B) UniFrac distance measures. Statistical significance of groupings was assessed by the analysis of similarities (ANOSIM), which test whether there is a significant difference between groups. Only in the case of the weighted Unifrac (**B**) we got a significant result for CO and OB groups (UNWEIGHTED: *p* = 0.175, R = 0.033; WEIGHTED: *p* = 0.039, R = 0.063), confirming that the variation between two main groups is not due to the type of taxa present in the microbiome but to their abundances.

**Figure 3 microorganisms-08-00485-f003:**
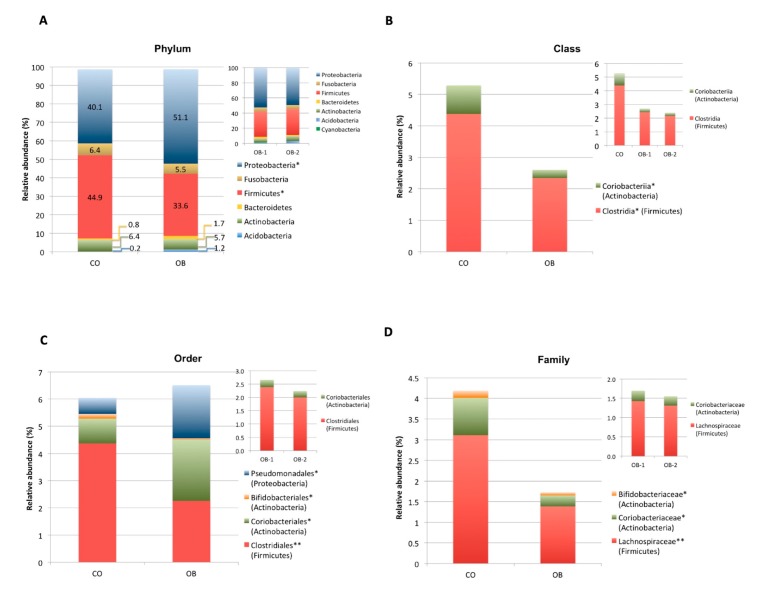
Composition analysis of gut microbiomes in the Control (CO) and Obese (OB) groups. The barplots show the relative abundance (%) of the 6 taxonomic levels from Phylum to Species, according to the SILVA database v.128. Each column in the plot represents a group, and each colour in the column represents the relative abundance (%) for each taxon. (**A**) Phyla having average abundance greater than 1% in at least one group of study were reported. Proteobacteria and Firmicutes were significant most and less abundant phyla, in obese respect to normal weight control group, respectively. (**B–F**): The barplots show the relative abundance (%) of taxonomic groups at class (**B**), order (**C**), family (**D**), genus (**E**) and species (**F**) levels which resulted significantly different among the two groups by Kruskal Wallis test. Not statistically significant difference in taxa abundance was observed when obese patients were divided according to obesity severity in OB-1 moderately obese and OB-2 severely obese groups (see upper right corner of panels A-E). * *p* < 0.05, ** *p* < 0.01.

**Figure 4 microorganisms-08-00485-f004:**
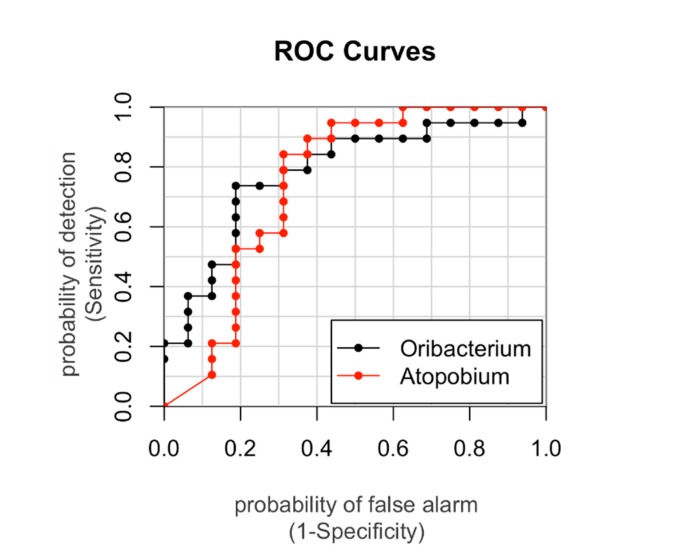
The areas under the Receiver operating characteristic curves (AUROCs) represent the specificity and sensitivity of the Amplicon Sequence Variants. The AUROC was calculated for those genera significantly different among the groups in order to identify those able to discriminate a specific group. Those assigned to Atopobium and Oribacterium had AUROCs of 75% and 78%, respectively. AUROC > 0.7 was considered suitable in discriminating with accuracy.

**Table 1 microorganisms-08-00485-t001:** Clinical and biochemical characteristics of obese patients.

	All obese pz (*n* = 19)		OB-1 (*n* = 13) (BMI 30–40)		OB-2 (*n* = 6) (BMI > 40)	
	Mean	SEM	Mean	SEM	Mean	SEM
Age, years	38.8	2.7	37.8	3.3	41.0	5.0
BMI *, kg/m^2^	39.3	1.4	36.0	0.8	46.5	2.0
Systolic blood pressure, mmHg	136	2.6	133.5	3.5	141.7	2.1
Diastolic blood pressure, mmHg	85.0 ^#^	80 – 90 ^#^	85.2	1.7	77.7	7.8
Heart rate, beats/min	82.2	1.7	83.4	1.9	79.7	3.4
Iron, μg/dL	91.1	6.8	97.0	8.3	78.3	11.3
Urea, mmol/L	34.2	1.7	33.7	2.5	35.3	1.5
Glucose *, mmol/L	5	0.1	4.8	0.1	5.5	0.2
Insulin, mIU/L	15.6	2.3	17.8	3.2	10.9	0.9
Creatinin, μmol/L	0.9 ^#^	0.8–1.0 ^#^	0.9 ^#^	0.8–1.0 ^#^	1.0	0.1
Total proteins, g/L	7.7	0.2	7.8	0.2	7.4	0.3
Albumin, g/L	4.3 ^#^	4.0–4.8 ^#^	4.3 ^#^	4.0–4.8 ^#^	4.3	0.3
Uric acid, mmol/L	6.1 ^#^	5.1–7.0 ^#^	6.1 ^#^	5.2–6.8 ^#^	6.3	0.7
Total bilirubin, μmol/L	0.7 ^#^	0.4–1.0 ^#^	0.7 ^#^	0.5–3.5 ^#^	0.7	0.1
Total cholesterol, mmol/L	5.3	0.2	5.1	0.2	5.6	0.3
Triglycerides, mmol/L	1.6	0.1	1.5	0.1	1.7	0.3
HDL-cholesterol, mmol/L	1.5	0.08	1.5	0.1	1.3	0.1
LDL-cholesterol, mmol/L	3.1	0.2	2.9	0.2	3.3	0.4
AST, U/L	23.6	2	24.5	2.7	21.7	2.8
ALT, U/L	28.9	5.4	31.0	7.7	24.5	4.6
ALP, U/L	55.6	3.6	56.9	5.3	52.8	1.9
GGT, U/L	26.7	3.6	31.0	4.7	17.3	2.0
Amylase, U/L	48.4	3.8	51.5	4.4	41.7	7.4

******p* < 0.001; ^#^ median value and 25th and 75th percentiles were reported for nonparametric distributions.

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
