# Peer review of "Characterization of the Duodenal Mucosal Microbiome in Obese Adult Subjects by 16S rRNA Sequencing"

_microorganisms, 2020, doi:10.3390/microorganisms8040485_

Round 1
Reviewer 1 Report
Nardelli et al. report about the characterization of the mucosa of obese adults with 16S sequencing. This manuscript is of certain interest, since mucosa of humans is normally not exactly easy to obtain, and this microbiome niche is not well represented by faeces. So this is a rather unique dataset.
While I in general like this whole setup, I have 2 main problems with this study:
- One drawback of the study is that the “healthy” controls were also suffering from a disease, which potentially impacts the microbiota (in general anything related to stomach pH apparently does, see any literature regarding microbiome and proton-pump-inhibitors; this normally also leads to an increase of oral bacteria in the gut, which is probably of interest for the discussion), and some conclusions are therefore not generalizable. But given that you can’t operate on actually healthy people, I think this is probably the best possible population you can get, and I think it’s a fine dataset. This issue should also be mentioned in the discussion
- - The authors do not report using negative controls (or positive ones; literature https://bmcbiol.biomedcentral.com/articles/10.1186/s12915-014-0087-z https://www.ncbi.nlm.nih.gov/pubmed/30497919 https://www.ncbi.nlm.nih.gov/pubmed/30997495 ). This is of critical importance, since the mucosal microbiome is a low biomass microbiome, and contaminations from any source could easily be mistaken as the real microbiome. It especially concerns me that the authors report a high amount of pseudomonales, which are common contaminants. It would be good to have the biom file as appendix (mentioned later), and if the authors would check if indeed common obvious contaminants (in general alphaproteobacteria, like rhizobia, ralstonia or methylobacterium, which are unlikely to be human associated) would be absent. Therefore I’m also concerned that Atopobium is mentioned, since it’s a common contaminant (see the list in the Eisenhofer article), and the other mentioned bacterium is an oral one, which could as well be a contamination from the processing. At best it would be if the authors had negative controls. Repeating a subsample now with negative controls will not work, since results are not comparable between runs and contamination varies between kit batches. If there is any doubt that there is more contamination, then I’d be sorry to say that this needs to be repeated (hopefully there are still biopsie specimen left over). But I truly hope that there are not more indications. This also needs to be in the discussion.
Comments:
- - Line 54: You forgot to cite https://www.ncbi.nlm.nih.gov/pmc/articles/PMC3779803/ . I also think that some important articles are missing from the intro, like https://www.ncbi.nlm.nih.gov/pmc/articles/PMC2677729/ or anything related to the HMP or MetaHit
- - Line 111-113: It is not mentioned if the biopsie specimen are in any ways treated, besides the DNA extraction. Was anything done to remove the human tissue, was the sample homogenized, anything?
- - Line 112+: So no negative and positive controls?
- - Line 153: Please specify exactly which database you used
- - Line 155/6: Versions of the packages, as well as R version itself need to be mentioned.
- - The statistical tests were performed in R?
- - Line 168: Version of the R packages
- - Line 172: Why was the student’s t-test used? I am pretty sure that this data is not normal distributed
- - The sequencing data obtained from all microbiome samples needs to be uploaded to the EBI/NCBI (not negotiable, this is the standard in the field)
- - It would also be good if the biom file itself could be added as supplementary file, to allow easier checking
- - Line 175+: I actually don’t see in the methods how the blood parameters were measured
- - Table 1: I would make the * more prominent (bold?), and it needs to be specified in the table description itself to which comparison the p value applies to. I am also seriously surprised that only glucose and BMI were different, given that it is mentioned that some participants had metabolic syndrome, and none of the cholesterol values and neither the insulin are statistically significantly different. Any explanation for this?
- - Figure 1: The capation mentions that the data was rarefied, but this is not mentioned in the methods. It is also in general not recommended, https://www.ncbi.nlm.nih.gov/pubmed/24699258
- - Figure 3: I really don’t like the colours, but that might be me.
- - Line 253-261: How many different ASVs were reported? I am asking because if only 3 different species were significantly different, that makes me wonder… with e.g. a 100 ASVs, with p<0.05, you’d expect by chance already 5 stat. Significant ASVs. So there actually should be more differences
- - I think the discussion is lacking some part comparing the duodenal mucosa with duodenal content and faeces
Author Response
Reviewer 1
Comments and Suggestions for Authors
Nardelli et al. report about the characterization of the mucosa of obese adults with 16S sequencing. This manuscript is of certain interest, since mucosa of humans is normally not exactly easy to obtain, and this microbiome niche is not well represented by faeces. So this is a rather unique dataset.
While I in general like this whole setup, I have 2 main problems with this study:
- One drawback of the study is that the “healthy” controls were also suffering from a disease, which potentially impacts the microbiota (in general anything related to stomach pH apparently does, see any literature regarding microbiome and proton-pump-inhibitors; this normally also leads to an increase of oral bacteria in the gut, which is probably of interest for the discussion), and some conclusions are therefore not generalizable. But given that you can’t operate on actually healthy people, I think this is probably the best possible population you can get, and I think it’s a fine dataset. This issue should also be mentioned in the discussion.As this Reviewer observed, it’s difficult to collect human biopsies from “healthy” individuals. We take the reviewer’s point that the microbiome of our “control” group, including lean subjects with clinical symptoms of gastroesophageal reflux (GORD), may present any differences with that of healthy individuals. Despite this, biopsies from our controls were taken before diagnosis and prospective GORD therapy with proton-pump-inhibitors (PPI). Further, multiple oral bacteria that were reported in gut microbiome of PPI-users included increased genera Rothia, Enterococcus, Streptococcus, Sthaphylococcus and species Escherichia Coli (Imhann, F et al. GUT, 2016;65;74048). Levels of these bacteria did not differ among our obese and control groups. This issue was added in the discussion.See Discussion: lines 354-361, page 11
- - The authors do not report using negative controls (or positive ones; literature https://bmcbiol.biomedcentral.com/articles/10.1186/s12915-014-0087-z https://www.ncbi.nlm.nih.gov/pubmed/30497919 https://www.ncbi.nlm.nih.gov/pubmed/30997495). This is of critical importance, since the mucosal microbiome is a low biomass microbiome, and contaminations from any source could easily be mistaken as the real microbiome. It especially concerns me that the authors report a high amount of pseudomonales, which are common contaminants. It would be good to have the biom file as appendix (mentioned later), and if the authors would check if indeed common obvious contaminants (in general alphaproteobacteria, like rhizobia, ralstonia or methylobacterium, which are unlikely to be human associated) would be absent. Therefore I’m also concerned that Atopobium is mentioned, since it’s a common contaminant (see the list in the Eisenhofer article), and the other mentioned bacterium is an oral one, which could as well be a contamination from the processing. At best it would be if the authors had negative controls. Repeating a subsample now with negative controls will not work, since results are not comparable between runs and contamination varies between kit batches. If there is any doubt that there is more contamination, then I’d be sorry to say that this needs to be repeated (hopefully there are still biopsie specimen left over). But I truly hope that there are not more indications. This also needs to be in the discussion. According to the recent review of Hornung BVH et al. (Hornung BVH et al. FEMS Microbiology Ecology, 95;fiz045, 2019), contamination should be considered at different steps in the microbiome analysis procedure.In order to reduce contamination’s risk, all biopsies were performed using standardized procedures. In particular, all endoscopes were reprocessed before use according to high-level disinfection protocols including manual washing, automated endoscope washer reprocessing and adequate drying/storage after rinsing (Multisociety guideline on reprocessing flexible gastrointestinal endoscopes: GASTROINTESTINAL ENDOSCOPY. 2011;73:1075-1084) defined as the destruction of all vegetative microorganisms, mycobacteria, small or nonlipid viruses, medium or lipid viruses, fungal spores and some, but not all, bacterial spores. All samples were taken with sterile biopsy forceps and the endoscopist wore sterile gloves and a mask. The samples used for the study were invariably taken before the other samples used for histological examination. Oral washing was performed for all patients before endoscopic study of enrolled patients: oral contamination can be considered the smallest possible, although unavoidable. After endoscopic biopsy, specimen was carefully extracted, without oral or gastroesophageal passage or contact. All single-use containers were used for all biopsies. All specimens were immediately placed in a sterile tube on dry ice and transferred to a –80°C freezer within 15 min and until DNA extraction. The same procedure was performed in the two different clinical centers which contributed equally to the enrolment of both obese and controls groups. We retain that this sampling procedure was effective in minimizing risk of contamination, either external either internal. All our samples (both from control and obese subjects) were processed concurrently in the same laboratory at Center of advanced Biotechnology CEINGE scarl, center that has implemented and maintains a quality management system which complies with the standard UNI EN ISO 9001:2015 for the diagnostic activities reported below, among which Genomix (next generation sequencing). The DNA extraction of all samples was performed in a pre-PCR designed room under a laminar-flow hood to protect laboratory staff, samples and experiments from contaminations. Pre-PCR wok was physically isolated from the post-PCR work. We extracted DNA from all samples by using the same QIAamp DNA mini kit and according manufacter’s instructions. The same kit was previously used by us and by others for duodenal microbiome studies in another disease model (Wacklin, P. et al,Inflamm Bowel Dis, 19;934-941, 2013; D’Argenio, V. et al, Am J Gastroenterol,111;879-890, 2016). DNA quantity was evaluated with the NanoDrop® ND-1000 UV-Vis spectrophotometer (NanoDrop Technologies, Wilmington, DE, USA). All DNA were diluted at 50 ng/µl for following PCR.Primer choice Finally, the primers were tested on primer blast (https://www.ncbi.nlm.nih.gov/tools/primer-blast/), using as reference the human genome, and no match were found, indicating no aspecific amplifications. Probe match analysis to verify taxa specificity was also carried out using both RDP database (https://rdp.cme.msu.edu), that found 1,188,968 hits within the domain “Bacteria”, and Silva TestPrime, that found 71% of match on Bacteria.PCR 16S rRNA The AMPure XP beads were used to purify the 16S V4 and V6 amplicon from contaminants (dNTPs, salts, primers, primer dimers).The quality assessment of PCR products was performed by TapeStation (Agilent Technologies, Santa Clara, California, USA). PCR product quantity was assessed by Qubit dsDNA BR assay (Thermo Fisher, Waltham, Massachusetts, USA) according to the manufacturer’s instructions. The average of 2 measurements was used to dilute the samples at 0,2 ng/µl for the next step.Index PCR Each amplicon of previous step was quantified by Qubit dsDNA HS assay (Thermo Fisher, Waltham, Massachusetts, USA) according to the manufacturer’s instructions. The average of 2 measurements was used to pool all the amplicons in equimolar amount.
- Comments:
- We also added the biom file in appendix
- See Discussion lines 362-375, pag 11
- Another limit of this study was the lack of use negative and/or positive controls in the DNA extraction and processing. In general, for good scientific practice, controls should be included at all steps in microbiome studies (Hornung,BVH, et al, FEMS Microboiology Ecology, 2019). In fact, as recently reported, DNA extraction kits and other laboratory reagents could be source of contamination (Salter, SJ et al, BMC Biology, 2014, 12:87). However, among the four commercially available tested and compared extraction kits by Salter et al., the QIAamp kit (the kit we used in this study) was one of the two less contaminated. Further, among contaminant taxa previously identified from multiple studies (Eisenhofer R, Trends in Microbiology, 2019, Salter, SJ et al, BMC Biology, 2014, 12:87; Glassing A. et al. Gut Pathog, 2016, 8,24 ), Ralstonia and Rhizobiales were virtually absent, and Methylbacterium was present in low relative abundance in both our groups of Control and Obese subjects (<0.2 and <0.3, respectively). Concerning Pseudomonadales and Atopobium, these taxa were present in all subjects but with opposite mean relative abundances in our two groups, that are Pseudomonadales was increased in obese and decreased in control subjects (1.94 and 0.58, respectively), whereas Atopobium was decreased in obese and increased in control subjects (0.26 and 0.88, respectively). Overall, our results are consistent with an insignificant bacterial contamination during sample processing.
- We also added in the Discussion the following paragraph.
- See Supplemental Material and methods
- We added all the above Sample Processing Procedures in the Supplementary Material
- The concentration of the pool was 4 nM; this pool was diluted to 8 pM for sequencing.
- Library Quantification, Normalization, and Pooling
- This step attaches dual indices and Illumina sequencing adapters using the Nextera XT Index Kit. This step was carried out in a post-PCR designed room under a PCR hood. No negative or positive controls were used. A step of purification (with AMPure XP beads to clean up the final library before quantification) and the quality assessment on Tape station were performed.
- PCR Purification and Quantitation
- The PCR was carried out in a PCR designed room under a PCR hood with HEPA-filtered vertical laminar flow to ensure the contamination control and PCR process repeatability. An integrated UV lamp enabled rapid decontamination of the work zone between experiments and prevents cross-contamination. Sterile water (Molecular Biology Grade Water, Corning) was used for reagent mix preparation and as negative control. During PCR reaction for 16S rRNA the negative control was amplified and no amplification was obtained on a 2% agarose gel indicating the absence of contamination.
- In this study, we used primers able to amplify the V4-V6 regions of the 16S rRNA. These primers were chosen after in-deep study of the literature. In particular: (i) the V4 region of 16S rRNA gene has been highly recommended as the gold standard for profiling of human gut microbiome by the MetaHIT consortium (Qin et al., Nature 2010; 464: 59-65; Lozupone et al., Genome Res 2013; 23: 1704-1714.); (ii) the V4-V5 hypervariable regions have shown to achieve a more accurate bacterial identification (Teng et al. Sci Rep. 2018; 8: 16321); and. (iii) the V4-V6 region has shown the best performance for gut microbiome profiling considering also the size of the amplicon (Yadav et al. DNA research 2019; 62: 147-156). In addition, the same primers have been already used by our group for other metagenomic studies (Iaffaldano et al. Sci Rep. 2018 Jul 23;8(1):11047; D’Argenio et al. Am J Gastroenterol. 2016 Jun;111(6):879-90; D’Argenio et al. Am J Gastroenterol. 2013 May;108(5):851-2).
- DNA extraction
- In this framework, the same researcher, reagents, and equipment were used to process all the samples of the present study. Personnel wore protective clothing and equipment to cover all exposed surfaces (i.e., disposable gloves, lab-coat, face-mask) and processed samples in controlled environment, as recently suggested (Eisenhofer,R. et al.Cell Press Reviews.Trends in Microbiology, 2019;27,105-17).
- See below attached Certificate N. 24277 for reviewer’s use.
- Sample processing
- See Supplemental Material and methods
- We added this Sampling Procedure in Supplementary Material
- Sampling (Collection of duodenal biopsies by endoscopy): in this step including an appropriate negative control is no easy task, particularly when the studied samples are faeces or biopsies. In fact, sampling the air or container with a different instrument could not be a true negative control since no material should be available for DNA extraction. Consequently, for this step we did not have negative controls, but our goal was to minimize technical variation as below reported.
- We take the Reviewer’s point about the possibility of contaminations in the microbiome studies.
- - Line 54: You forgot to cite https://www.ncbi.nlm.nih.gov/pmc/articles/PMC3779803/ . I also think that some important articles are missing from the intro, like https://www.ncbi.nlm.nih.gov/pmc/articles/PMC2677729/ or anything related to the HMP or MetaHit
- We added the new references to the IntroductionSee Introduction lines 54 and 67, pag.2 and References 10 and 17
- - Line 111-113: It is not mentioned if the biopsie specimen are in any ways treated, besides the DNA extraction. Was anything done to remove the human tissue, was the sample homogenized, anything?
- DNA was extracted from biopsies without any preliminary treatment
- - Line 112+: So no negative and positive controls?No positive control in the sample processing was used.
- See Supplemental Material and methods
- We added the detailed Sampling Procedure in Supplementary Material
- During PCR reaction for 16S rRNA a negative control (sterile water) was amplified and no amplification was obtained on a 2% agarose gel supporting the absence of contamination.
- We used the silva database version 128 formatted for DADA2 software and available at the link:https://zenodo.org/record/824551#.XmIcO5NKhuU.See lines 161,162, pag.4· - Line 155/6: Versions of the packages, as well as R version itself need to be mentioned.See line 164, pag.4Yes, we added this information in the paper.· - Line 168: Version of the R packages See line 180, pag.4
- We added R version.
- See lines 176-177, pag. 4
- · - The statistical tests were performed in R?
- We added versions of the packages as requested.
- We added the link in the manuscript as suggested by the reviewer.
- · - Line 153: Please specify exactly which database you used
- - Line 172: Why was the student’s t-test used? I am pretty sure that this data is not normal distributed.We thank the Reviewer for his observation.
- See Statistical Analysis pag. 4 and Table 1.
- The parameters investigated were expressed as mean and standard error of the mean (SEM) (parametric distributions) or as the median value and 25th and 75th percentiles (nonparametric distributions). The Student’s ‘t’ and Mann–Whitney tests were used to compare parametric and nonparametric data, respectively. P values < 0.05 were considered statistically significant. Correlation analysis was performed with the SPSS package for Windows (ver. 18; SPSS, Inc.). We modified the text and the Table 1 accordingly.
- - The sequencing data obtained from all microbiome samples needs to be uploaded to the EBI/NCBI (not negotiable, this is the standard in the field)16S sequencing raw data (FASTQ) obtained from all microbiome samples have been submitted to the NCBI SRA repository under the accession number PRJNA611696 and will be available after paper acceptance.
- See lines 136, 137 - pag. 3
- In the meanwhile we provide the link for the reviewers to allow them seeing and checking the submission: https://dataview.ncbi.nlm.nih.gov/object/PRJNA611696?reviewer=kk5ojd89s1ajt7br6u32u8dhb0
- - It would also be good if the biom file itself could be added as supplementary file, to allow easier checkingWe added the BIOM FILE as appendix for reviewer’s use.
- - Line 175+: I actually don’t see in the methods how the blood parameters were measuredWe added this information in the text.See Sample collection and Biochemical investigations paragraph, lines 112, 113, pag. 3.
- - Table 1: I would make the * more prominent (bold?), and it needs to be specified in the table description itself to which comparison the p value applies to. I am also seriously surprised that only glucose and BMI were different, given that it is mentioned that some participants had metabolic syndrome, and none of the cholesterol values and neither the insulin are statistically significantly different. Any explanation for this?We modified the Table 1 as suggested by the Reviewer.See Table 1Metabolic syndrome was diagnosed in 6/19 obese patients considering the combination of three out of five risk factors according to AHA criteria (Grundy et al., 2005). Below are the risk factors of each patient. No one of our participants with MS had high levels of cholesterol likely for their young age.
|
Sample |
BMI (>30 kg/m2) |
Pressure (>135/85 mmHg) |
Triglycerides (>1.69 mmol/L) |
Glucose (>6.1 mmol/L) |
HDL Cholesterol mmol/L (<40 males and <50 females) |
|
MS1 |
X |
X |
X |
|
|
|
MS2 |
X |
X |
X |
|
|
|
MS3 |
X |
X |
X |
|
|
|
MS4 |
X |
X |
|
X |
|
|
MS5 |
X |
X |
X |
|
|
|
MS6 |
X |
X |
|
|
X |
- - Figure 1: The capation mentions that the data was rarefied, but this is not mentioned in the methods. It is also in general not recommended, https://www.ncbi.nlm.nih.gov/pubmed/24699258We thank the Reviewer for the right suggestion and, in order to follow it, we have recomputed the alpha and beta diversity. In the case of alpha diversity there’s no need to normalize data at all, as suggested by Phyloseq FAQ section (https://www.bioconductor.org/packages/devel/bioc/vignettes/phyloseq/inst/doc/phyloseq-FAQ.html). In the case of the beta diversity we applied the variance stabilizing transformation, through DEseq2 package, as suggested by the paper cited by the reviewer. The new plots, statistics and text have been added to the manuscript.
- See lines 169-174, pag. 4, Figure 1 and Figure 1 legend
- - Figure 3: I really don’t like the colours, but that might be me.We took in consideration the reviewer’s opinion and changed the colours accordingly.
- See Figure 3
- - Line 253-261: How many different ASVs were reported? I am asking because if only 3 different species were significantly different, that makes me wonder… with e.g. a 100 ASVs, with p<0.05, you’d expect by chance already 5 stat. Significant ASVs. So there actually should be more differencesThe single ASVs have been annotated by SILVA database and the counts corresponding to different ASVs belonging to the same taxa (for each taxonomy level) have been collapsed by summing them. This means that 3 significative species correspond to much more ASVs.
- - I think the discussion is lacking some part comparing the duodenal mucosa with duodenal content and faeces.We take the reviewer’s point and added a new paragraph in the discussion relative to the comparison of duodenal mucosa microbiome with those of duodenal content and faeces.
- See Discussion, lines 336-347, pag.10
Reviewer 2 Report
The manuscript by Nardelli et al. describes the study of duodenal microbiome changes, associated with obesity. To date, due to obvious sampling difficulties, that is one of a few studies, focusing on obesity-induced alteration of microbiota in small intestine, as opposed to the classical approach, based on fecal samples. Due to this fact work of Nardelli et al. looks very advantageous and might have a significant contribution for the further development of microbiota-correction techniques and treatments. Nevertheless, quality of the presentation is seriously compromised due to the number of methodological disadvantages:
(lane 113) current techniques for metagenomic DNA extraction includes bead-beating step to achieve uniform efficiency of DNA extraction from different taxa. Some kind of spike-in control must be used to persuade reader, that results reflect the real composition of duodenal microbial community. Nevertheless, if all the biomass was used for DNA extraction, authors should provide reference for the validation of QIAamp DNA mini kit for the analysis of intestine samples.
(lane 113) due to low amount of biomass in biopsy samples, serious biases might be induced by contamination of reagents (Salter et al., 2014), so negative controls or sample dilutions must be sequenced.
(lanes 119 – 122) authors should present an argument for primer choice. It might be either a reference on the study, where exactly these primers were utilized, or a detailed analysis of taxa specificity, performed with PrimerProspector (http://pprospector.sourceforge.net/) or Silva TestPrime web tool (https://www.arb-silva.de/search/testprime/).
(lanes 123-125) Nextera XT is not sequencing, but library prep protocol. Sentence must be rephrased.
(lanes 137-146) authors describe the experimental process of data processing methods. All the results must be presented in Supplementary
(lanes 137-146) nonstandard option for the data processing must be validated. Best option there should be sequencing and further analysis of mock community.
(lane 148) meaning of DADA2 parameters should be explained
(lanes 137-148) There are tools, allowing analysis of non-overlapping read pairs, for example IM-TORNADO (Jeraldo et al, 2014).
(lanes 196-198) analysis of sup. table 1 shows that over 60% of sequence reads were removed after filtering. Was the sequencing run so poor? If yes – are the obtained results reliable?
(Figure 2B) circles drawn on the top of PCA graph looks very forced. Real position of data points on the graph does not allow such grouping.
(Results) one of well-known obesity markers is Firmicutes to Bacteroidetes ratio. Was it checked for duodenal samples?
(Discussion) – results, especially the over-representation of mentioned taxa in obese patients, must be compared with data, obtained by Angelakis et al (2015).
(Whole manuscript). English must be significantly improved.
Due to these facts, despite I consider that this study is very interesting and important, I would suggest a major revision of this manuscript.
Author Response
Reviewer 2
Comments and Suggestions for Authors
The manuscript by Nardelli et al. describes the study of duodenal microbiome changes, associated with obesity. To date, due to obvious sampling difficulties, that is one of a few studies, focusing on obesity-induced alteration of microbiota in small intestine, as opposed to the classical approach, based on fecal samples. Due to this fact work of Nardelli et al. looks very advantageous and might have a significant contribution for the further development of microbiota-correction techniques and treatments. Nevertheless, quality of the presentation is seriously compromised due to the number of methodological disadvantages:
1 (lane 113) current techniques for metagenomic DNA extraction includes bead-beating step to achieve uniform efficiency of DNA extraction from different taxa. Some kind of spike-in control must be used to persuade reader, that results reflect the real composition of duodenal microbial community. Nevertheless, if all the biomass was used for DNA extraction, authors should provide reference for the validation of QIAamp DNA mini kit for the analysis of intestine samples.
We used all microbial biomass for DNA extraction from our duodenal samples. The QIAamp DNA mini kit for the analysis of intestine samples was previously used by us and others (Wacklin, P. et al,Inflamm Bow Dis, 19;934-941, 2013; D’Argenio, V. et al, Am J Gastroenterol,111;879-890, 2016)
2 (lane 113) due to low amount of biomass in biopsy samples, serious biases might be induced by contamination of reagents (Salter et al., 2014), so negative controls or sample dilutions must be sequenced.
We take the Reviewer’s point about the possibility of contaminations in the microbiome studies.
According to the recent review of Hornung BVH et al. (Hornung BVH et al. FEMS Microbiology Ecology, 95;fiz045, 2019), contamination should be considered at different steps in the microbiome analysis procedure. Below are detailed our procedural steps:
Sample processing
All our samples (both from control and obese subjects) were processed concurrently in the same laboratory at Center of advanced Biotechnology CEINGE scarl, center that has implemented and maintains a quality management system which complies with the standard UNI EN ISO 9001:2015 for the diagnostic activities reported below, among which Genomix (next generation sequencing).
See below attached Certificate N.24277 for reviewer’s use.
In this framework, the same researcher, reagents, and equipment were used to process all the samples of the present study. Personnel wore protective clothing and equipment to cover all exposed surfaces (i.e., disposable gloves, lab-coat, face-mask) and processed samples in controlled environment, as recently suggested (Eisenhofer,R. et al.Cell Press Reviews.Trends in Microbiology, 2019;27,105-17).
DNA extraction
The DNA extraction of all samples was performed in a pre-PCR designed room under a laminar-flow hood to protect laboratory staff, samples and experiments from contaminations. Pre-PCR wok was physically isolated from the post-PCR work. We extracted DNA from all samples by using the same QIAamp DNA mini kit and according manufacter’s instructions. DNA quantity was evaluated with the NanoDrop® ND-1000 UV-Vis spectrophotometer (NanoDrop Technologies, Wilmington, DE, USA). All DNA were diluted at 50 ng/ml for following PCR.
PCR 16S rRNA
The PCR was carried out in a PCR designed room under a PCR hood with HEPA-filtered vertical laminar flow to ensure the contamination control and PCR process repeatability. An integrated UV lamp enabled rapid decontamination of the work zone between experiments and prevents cross-contamination. Sterile water (Molecular Biology Grade Water, Corning) was used for reagent mix preparation and as negative control. During PCR reaction for 16S rRNA the negative control was amplified and no amplification was obtained on a 2% agarose gel indicating the absence of contamination.
PCR Purification and Quantitation
The AMPure XP beads were used to purify the 16S V4 and V6 amplicon from contaminants (dNTPs, salts, primers, primer dimers).The quality assessment of PCR products was performed by TapeStation (Agilent Technologies, Santa Clara, California, USA). PCR product quantity was assessed by Qubit dsDNA BR assay (Thermo Fisher, Waltham, Massachusetts, USA) according to the manufacturer’s instructions. The average of 2 measurements was used to dilute the samples at 0,2 ng/µl for the next step.
Index PCR
This step attaches dual indices and Illumina sequencing adapters using the Nextera XT Index Kit. This step was carried out in a post-PCR designed room under a PCR hood. No negative or positive controls were used. A step of purification (with AMPure XP beads to clean up the final library before quantification) and the quality assessment on Tape station were performed.
Library Quantification, Normalization, and Pooling
Each amplicon of previous step was quantified by Qubit dsDNA HS assay (Thermo Fisher, Waltham, Massachusetts, USA) according to the manufacturer’s instructions. The average of 2 measurements was used to pool all the amplicons in equimolar amount.
The concentration of the pool was 4 nM; this pool was diluted to 8 pM for sequencing.
We added the Sample Processing Procedures in the Supplementary Material
See Supplemental Material and methods
We also added in the Discussion the following paragraph.
Another limit of this study was the lack of use negative and/or positive controls in the DNA extraction and processing. In general, for good scientific practice, controls should be included at all steps in microbiome studies (Hornung,BVH, et al, FEMS Microboiology Ecology, 2019). In fact, as recently reported, DNA extraction kits and other laboratory reagents could be source of contamination (Salter, SJ et al, BMC Biology, 2014, 12:87). However, among the four commercially available tested and compared extraction kits by Salter et al., the QIAamp kit (the kit we used in this study) was one of the two less contaminated. Further, among contaminant taxa previously identified from multiple studies (Eisenhofer R, Trends in Microbiology, 2019, Salter, SJ et al, BMC Biology, 2014, 12:87; Glassing A. et al. Gut Pathog, 2016, 8,24), Ralstonia and Rhizobiales were virtually absent, and Methylbacterium was present in low relative abundance in both our groups of Control and Obese subjects (<0.2 and <0.3, respectively). Concerning Pseudomonadales and Atopobium, these taxa were present in all subjects but with opposite mean relative abundances in our two groups, that are Pseudomonadales was increased in obese and decreased in control subjects (1.94 and 0.58, respectively), whereas Atopobium was decreased in obese and increased in control subjects (0.26 and 0.88, respectively). Overall, our results are consistent with an insignificant bacterial contamination during sample processing.
See Discussion lines 362-375, pag 11
2 (lanes 119 – 122) authors should present an argument for primer choice. It might be either a reference on the study, where exactly these primers were utilized, or a detailed analysis of taxa specificity, performed with PrimerProspector (http://pprospector.sourceforge.net/) or Silva TestPrime web tool (https://www.arb-silva.de/search/testprime/).
In this study, we used primers able to amplify the V4-V6 regions of the 16S rRNA. These primers were chosen after in-deep study of the literature. In particular: (i) the V4 region of 16S rRNA gene has been highly recommended as the gold standard for profiling of human gut microbiome by the MetaHIT consortium (Qin et al., Nature 2010; 464: 59-65; Lozupone et al., Genome Res 2013; 23: 1704-1714.); (ii) the V4-V5 hypervariable regions have shown to achieve a more accurate bacterial identification (Teng et al. Sci Rep. 2018; 8: 16321); and. (iii) the V4-V6 region has shown the best performance for gut microbiome profiling considering also the size of the amplicon (Yadav et al. DNA research 2019; 62: 147-156). In addition, the same primers have been already used by our group for other metagenomic studies (Iaffaldano et al. Sci Rep. 2018 Jul 23;8(1):11047; D’Argenio et al. Am J Gastroenterol. 2016 Jun;111(6):879-90; D’Argenio et al. Am J Gastroenterol. 2013 May;108(5):851-2).
Finally, the primers were tested on primer blast (https://www.ncbi.nlm.nih.gov/tools/primer-blast/), using as reference the human genome, and no match were found, indicating no aspecific amplifications. Probe match analysis to verify taxa specificity was also carried out using both RDP database (https://rdp.cme.msu.edu), that found 1,188,968 hits within the domain “Bacteria”, and Silva TestPrime, that found 71% of match on Bacteria.
This has been now specified in the revised manuscript under Supplemental Materials and Methods file.
See Supplemental Materials and Methods
3 (lanes 123-125) Nextera XT is not sequencing, but library prep protocol. Sentence must be rephrased.
Thanks, we have rephrased the sentence.
See lines 126-127, pag.3
4 (lanes 137-146) authors describe the experimental process of data processing methods. All the results must be presented in Supplementary
The strategy of using only the forward reads in case of scarcely overlapping paired ends is a very common option and is often suggested by the software developer as alternative to the concatenation (as example please see https://www.drive5.com/usearch/manual/merge_badrev.html). The V4 region is known to be very sensitive as marker for bacterial and phylogenetic analysis (Yang, Bo, Yong Wang, and Pei-Yuan Qian. "Sensitivity and correlation of hypervariable regions in 16S rRNA genes in phylogenetic analysis." BMC bioinformatics 17.1 (2016): 135.). Furthermore, our results are confirmed when the R2 reads are added, as in the case of combining paired (obtained by qiime join_paired_ends.py), R1 and R2 reads. This strategy is very similar to that used by the software allowing the analysis of non-overlapping reads, like IM-TORNADO, discussed in the next answer. The Venn diagram shows the intersection of the significative genera obtained adding the reverse reads (join+R1+R2) and using only the forward reads (R1). Only one more significative genus is obtained: Megasphaera (p: 0.046), but given the lack of consensus we decided to not consider it.
Common elements in DADA2_R1 and join+R1+R2:
Oribacterium
Atopobium
Stomatobaculum
Bifidobacterium
See lines 146-150 pag 4 and Supplementary Data Processing
5 (lanes 137-146) nonstandard option for the data processing must be validated. Best option there should be sequencing and further analysis of mock community.
The performance of different kits on varying mock communities has not yet been tested (Hornung,VBH, 2019), therefore extracting and sequencing DNA from one mock community does not necessarily guarantee the suitability of the procedure for real biological samples.
Supplemental Data Processing
6 (lane 148) meaning of DADA2 parameters should be explained
We have described the meaning of DADA2 parameters as requested.
See lines 151-158, pag 4
7 (lanes 137-148) There are tools, allowing analysis of non-overlapping read pairs, for example IM-TORNADO (Jeraldo et al, 2014).
IM-TORNADO (Jeraldo et al. 2014), treat the forward and reverse reads separately and pair what is possible to pair. The pipeline then goes in parallel for the three different input files (R1, R2 and paired) and provide three different results. We think that since the R1 and R2 reads map to two different regions is not recommendable to use them separately. Furthermore, the V4 region is much more sensitive than the V6 (Yang, Bo, Yong Wang, and Pei-Yuan Qian. "Sensitivity and correlation of hypervariable regions in 16S rRNA genes in phylogenetic analysis." BMC bioinformatics 17.1 (2016): 135.), as discussed above. Furthermore DADA2 has been chosen given its high accuracy and the capability to use ASVs rather than OTUs, whose advantages are well described in https://www.nature.com/articles/ismej2017119. Nonetheless, to take into account the reviewer’s suggestion, we have used IM-tornado on our data. The three output biom files provided by the software have been imported into phyloseq objects in R environment and the relative abundances differences at genus level analyzed by Kruskal-Wallis test. As shown by the first Venn diagram attached down, there is a scarce overlap of the results obtained by tornado_R1, tornado_R2 and tornado paired (tornado_p) reads, just 2, which are Atopobium and Oribacterium. The second Venn diagram shows that if we add both the results obtained by our alternative approach based on combining joined and unjoined reads and by the approach shown in this study (R1 (our)) still only 2 remains, Oribacterium and Atopobium. Stomatobaculum is not present in rdp9 database used by IM-Tornado software, which contain much less taxa than Silva db (10172 vs 190661). This comparison of results obtained by different approaches highlights that our strategy was definitely more conservative and that the genera we found as discriminative, namely Oribacterium and Atopobium, are confirmed by all the analyses.
8 (lanes 196-198) analysis of sup. table 1 shows that over 60% of sequence reads were removed after filtering. Was the sequencing run so poor? If yes – are the obtained results reliable?
We agree with the reviewer that we lost a high percentage of reads during the filtering steps. We have used very stringent quality parameters on purpose and used DADA2 for its high accuracy, as demonstrated by the authors (Benjamin J., et al. "DADA2: high-resolution sample inference from Illumina amplicon data." Nature methods 13.7 (2016): 581; Callahan, Benjamin J., et al. "High-throughput amplicon sequencing of the full-length 16S rRNA gene with single-nucleotide resolution." Nucleic acids research 47.18 (2019): e103-e103.). This choice was guided by the fact that we decided to use only the forward reads and then an amplicon shorter than the expected one. If this strategy would have been associated to mild quality filters maybe our results could be less reliable due to lower specificity. In our case we used only high quality reads and applied several filtering steps (quality, length, error rates, denoising, chimeras) to ensure the reliability of the results. Anyway, in our opinion, the number of reads remaining at the end of the filters are still enough for a good and complete analysis.
9 (Figure 2B) circles drawn on the top of PCA graph looks very forced. Real position of data points on the graph does not allow such grouping.
Due to the significance of ANOSIM test we thought to ease the grouping view, but we accept the reviewer’s comment and removed the circles.
See Figure 2.
10 (Results) one of well-known obesity markers is Firmicutes to Bacteroidetes ratio. Was it checked for duodenal samples?
The microbial Firmicutes/Bacteroidetes ratio is reported to be increased in obesity, but it is usually calculated in stool samples, whose microbiome is mostly representative of colon (George Kunnackal Jhonn and Gerard E. Mullin, Curr Oncol Rep , 2016, 18:45). The latter microbiome is largely different from those present in other intestinal tracts such as duodenum or jejunum, which host lower than colon abundances of Bacteroidetes, probably related to a limited availability of mucin as carbon source for Bacteroidetes (Angelakis et al 2015, Wang et al,2005).
We added this comment to the discussion.
See Discussion, lines 348-352, pag. 11
11 (Discussion) – results, especially the over-representation of mentioned taxa in obese patients, must be compared with data, obtained by Angelakis et al (2015).
We added a new paragraph in the discussion comparing our data with those of Angelakis et al (2015).
See Discussion, lines 336-347, pag 10
12 (Whole manuscript). English must be significantly improved.
English in the whole manuscript was revised.
Due to these facts, despite I consider that this study is very interesting and important, I would suggest a major revision of this manuscript.
Round 2
Reviewer 1 Report
I'm quite happy with the response of the authors. It seems that they put a good deal of thought into the response, and also before already during the design of their experiments. It's nice to see :).
I only have a few small comments, otherwise I don't see any issues.
- My bad, could you add the .tsv version of the biom file to the supplementary? That's what I originally meant.
- Figure 1: What's with the caption "severity"?
- Line 303: Short chain fatty acid producers
- Line 366/7: "the QIAamp kit (the kit we used in
this study) was one of the two less contaminated. " a) less contaminated kits. b) please add a sentence saying that this can in general not be taken as an indication, since this varies by batch (or remove this part).
Also good luck with your future research, and I hope the virus doesn't hit you people down in the south that bad as in the north.
Author Response
Reviewer 1: Comments and Suggestions for Authors
I'm quite happy with the response of the authors. It seems that they put a good deal of thought into the response, and also before already during the design of their experiments. It's nice to see :).
I only have a few small comments, otherwise I don't see any issues.
- My bad, could you add the .tsv version of the biom file to the supplementary? That's what I originally meant.
As suggested we attached in Supplemental material the .tsv version of the biom file.
See Supplemental .tsv version biom file
- Figure 1: What's with the caption "severity"?
Thanks for the note, we now removed “severity” in Figure 1.
See Figure 1
- Line 303: Short chain fatty acid producers
We modified Line 305 as suggested
See pag, 10 Line 305
- Line 366/7: "the QIAamp kit (the kit we used in
this study) was one of the two less contaminated. " a) less contaminated kits. b) please add a sentence saying that this can in general not be taken as an indication, since this varies by batch (or remove this part).
As reviewer 1 suggested we reformulated the sentence.
See pag, 11 lines 367-368
Also good luck with your future research, and I hope the virus doesn't hit you people down in the south that bad as in the north.
We wish to thank this reviewer for his helpful suggestions and for the concern for the health of the Southern Italy people. Hopefully the Corona virus outbreak will soon be wiped out worldwide.

Reviewer 2 Report
Dear colleagues,
Thank you for your detailed responses. It clarifies a lot and changes you've made significantly improve the manuscript. For my opinion it is ready for publication.
All the best!
Author Response
We thank the Reviewer 2.